# The Transcription Machinery and the Driving Force of the Transcriptional Molecular Condensate: The Role of Phosphates

**DOI:** 10.3390/cimb47070571

**Published:** 2025-07-20

**Authors:** Raúl Riera Aroche, Esli C. Sánchez Moreno, Yveth M. Ortiz García, Andrea C. Machado Sulbarán, Lizbeth Riera Leal, Luis R. Olivas Román, Annie Riera Leal

**Affiliations:** 1Department of Research in Physics, University of Sonora, Hermosillo 83000, Mexico; rriera@cifus.uson.mx; 2Research and Higher Education Center of UNEPROP, Hermosillo 83105, Mexico; eslicamila@gmail.com (E.C.S.M.); yveth.ortiz@academicos.udg.mx (Y.M.O.G.); rieraleal@yahoo.com.mx (L.R.L.); 3Dermatology Department, General Hospital of the State of Sonora, Hermosillo 83000, Mexico; 4Research Institute of Dentistry, Center of Health Sciences, University of Guadalajara, Guadalajara 44100, Mexico; 5Childhood and Adolescence Cancer Research Institute, Center of Health Sciences, University of Guadalajara, Guadalajara 44100, Mexico; andrecaroms@gmail.com; 6Dermatology Department, Ayala Hospital-HGR 45 IMSS, Guadalajara 44100, Mexico; 7Pathology Department, General Hospital of the State of Sonora, Hermosillo 83000, Mexico; luis_chaparro@hotmail.com

**Keywords:** phosphorylation, transcription processes, hydrogen proton transfer, protein–DNA–water condensate

## Abstract

The dynamic phosphorylation of the human RNA Pol II CTD establishes a code applicable to all eukaryotic transcription processes. However, the ability of these specific post-translational modifications to convey molecular signals through structural changes remains unclear. We previously explained that each gene can be modeled as a combination of n circuits connected in parallel. RNA Pol II accesses these circuits and, through a series of pulses, matches the resonance frequency of the DNA qubits, enabling it to extract genetic information and quantum teleport it. Negatively charged phosphates react under RNA Pol II catalysis, which increases the electron density on the deoxyribose acceptor carbon (2’C in the DNA sugar backbone). The phosphorylation effect on the stability of a carbon radical connects tyrosine to the nitrogenous base, while the subsequent pulses link the protein to molecular water through hydrogen bonds. The selective activation of inert C(sp^3^)–H bonds begins by reading the quantum information stored in the nitrogenous bases. The coupling of hydrogen proton transfer with electron transfer in water generates a supercurrent, which is explained by the correlation of pairs of the same type of fermions exchanging a boson. All these changes lead to the formation of a molecular protein–DNA–water transcriptional condensate.

## 1. Introduction

Phosphate esters and anhydrides dominate the living world. The main reservoirs of biochemical energy—adenosine triphosphate (ATP), creatine phosphate, and phosphoenolpyruvate—are phosphates [1]. Unlike synthetic organic chemistry, nature has favored phosphates, as the leaving group is typically phosphate or pyrophosphate (PP_i_) in most natural biochemical and metabolic reactions. No other residue appears to fulfill the numerous roles of phosphate in biochemistry, particularly as an energy source. The importance of protein phosphorylation in life is evident in nearly every process. The human proteome contains around 100,000 potential phosphorylation sites [2]. In cellular signaling, one-third of all proteins in the cell are phosphorylated at any given moment [3,4].

While the hydrophobic nitrogenous bases form complementary Hydrogen (H)-bonded pairs (A-T and G-C) within the macromolecule to minimize contact with water molecules, negatively charged phosphate groups in the backbone remain exposed to the solution [5]. Under physiological conditions, water molecules surrounding DNA create a dynamic network of H-bonds that interact with the structure and dynamics of the double helix. DNA maintains a specific B-form with a hydration shell of 30–50 water molecules per nucleotide pair [6]. These water molecules establish a dynamic network of H-bonds akin to bulk water. Water molecules that interact directly with the DNA atoms make up the most structured part of the hydration shell, particularly within the grooves of the double helix; these water complexes are the most ordered [7,8]. Specifically, within the DNA minor groove, a single water molecule can bridge two atoms of different nucleotide bases, forming a chain of structured water molecules known as the DNA hydration spine [6,9].

Here, after characterizing the transcription process and DNA as a quantum computer [10,11], we explain why phosphates serve as the pulse emitted by RNA polymerase II to induce decoherence in the system and transfer genetic information. Additionally, while discussions continue about the formation of molecular condensates during transcription, this article, for the first time, establishes the relationship between the proton released during the phosphorylation series of pulses, conductivity, and the structural formation of water associated with DNA, enabling the system to function in a compartmentalized manner. The relationship between amino acids and a water molecule, a strong structural link with the substrate, has been well documented. For example, solid theoretical evidence indicates that the conserved aspartate acts as a proton acceptor in some reaction steps. The conserved features of the active site—including conserved amino acid residues, Mg^2+^ ions, and water—are likely crucial for stabilizing the transition state and regulating the timing of the phosphoryl transfer reaction [12].

H^+^ ions move through transient formations of hydronium ions produced by water protonation. In some studies, the maximum ionic conductivity of the membranes increases considerably when saturated with water [13]. Micro-water channels facilitate proton conduction and transition, thereby enhancing the proton conductivity of the membranes [14]. However, what remains unexplained is how electrical conductivity is physically achieved. The proton jump is frequently considered to cause a proton current.

Using valence theory and orbital analysis, we demonstrate how the proton jump occurs and dismiss the idea of a pure proton current, as suggested in earlier studies. We introduce the concept of a supercurrent of electrons driven by proton movement and propose the corresponding physical-mathematical model. The identification of resonant quantum states where electron pairs exchange bosons is a novel physical model, initially explored by our group to explain aromaticity in benzene [10,15] and, in this work, to elucidate the superconductivity of water during transcription. This leads us to suggest that, beyond their previously described qualities, phosphates play a crucial role as the pulse, energy transfer agents, and sources of protons necessary for the formation of molecular condensates—without which transcription cannot occur.

The proper function of the genome depends on the spatial organization of DNA. The assembly of the Pol II complex occurs at the gene promoter [16,17,18]. Protein compartmentalization, driven by multivalent interactions and liquid–liquid phase separation, provides a mechanism to regulate protein functions spatially [19,20,21,22,23]. An emerging model for the transcription of protein-coding genes suggests that distinct transient condensates form at gene promoters and within gene bodies, concentrating the factors necessary for each transcription step. RNA pol II may shuttle between these condensates in a phosphorylation-dependent manner [24,25,26].

RNA polymerase II (Pol II)-driven transcription is a unidirectional multistep process. Each cycle can be divided into phases: initiation, elongation, and termination. Thousands of unique transcription cycles occur with the concerted action of transcriptional cyclin-dependent kinases, their cognate cyclins, and the opposing activity of transcriptional phosphatases [11]. The dynamic phosphorylation of multiple residues within the Pol II complex carboxy-terminal domain (CTD) of the RPB1 subunit is central to controlling Pol II-driven transcription. In humans, this region comprises 52 heptad repeats of Tyrosine (Tyr), Serine (Ser), Proline (Pro), and Threonine (Thr) (Tyr_1_Ser_2_Pro_3_Thr_4_Ser_5_Pro_6_Ser_7_) [27]. A growing list of proteins specifically recognizes post-translationally modified CTD residues, integrating additional transcriptional and co-transcriptional activities. DNA-binding transcriptional regulators are the most essential components of these regulatory systems [28].

We recently described a quantum model of transcription. In this work, every gene in DNA was modeled as a combination of 
n
-circuits using quantum electrodynamics [11], based on establishing DNA qubits (with the central H-bond acting like a Josephson junction) as Radio Frequency Superconducting Quantum Interference Devices (RF SQUIDs) with intrinsically quantum behavior and the sugar-phosphate backbone as a finite transmission line with distributed parameters (Figure 1A). DNA is unwound in the transcription bubble condensate before the RNA pol II catalytic site. The growing RNA is attached to the active center cleft with its 3′ end, forming the nine-bp hybrid duplex with the DNA template strand. The high elongation complex stability depends on the tight DNA–RNA binding, which facilitates transcription processivity [29].

RNA Pol II changes the external magnetic flux through a phosphorylation mechanism with the exact pulse parameters 
n
 times to read the DNA qubits. This generates a phase difference. A pulse is a finite time series of complex values (a pulse amplitude at a specific time). However, a pulse was defined in DNA as a finite time series of phosphorylation reactions [11]. In our work, we connect the structure of water associated with DNA to the phosphorylation process through proton release. This occurs when the phosphoryl group of ATP is transferred to serine-threonine and tyrosine residues in the CTD. We introduce the concept of a supercurrent of electrons induced by the movement of a proton and propose a physical-mathematical model. Biomolecular condensates formed through the phase separation of IDPs/IDRs and nucleic acids contain water. Water accounts for about 60–70% of the condensate volume and is believed to affect interactions between chains and the surrounding solvent, influencing the dynamics of the condensates [6].

## 2. Materials and Methods

This work presents a theoretical approach to describe the role of phosphates during the transcription cycles using concepts from physics and chemistry. The outline of this paper is as follows: Section 3.1.1 introduces phosphates as driving forces for RNA Pol II. Section 3.1.2 presents the theoretical model of phosphorylation-inducing interactions between RNA Pol II and DNA. Section 3.2 highlights the role of water molecules in DNA and intrinsically disordered proteins (IDPs). Section 3.2.1 proposes a physical model of phosphorylation mediating H^+^ proton transfer to explain superconductivity in water in Section 3.2.2 and Section 3.2.3. Section 4 discusses the theoretical background that underlies our model definition. All figures were created and edited using Adobe Illustrator (AI) software 2025. For the mathematical analysis, we used the Word Office 365 Equation Editor.

## 3. Results

### 3.1. Characterization of the Pulse During the Transcription Process

#### 3.1.1. Phosphates as the Driving Force

The dynamic phosphorylation of Tyr-Ser-Thr-Pro in the Pol II CTD establishes a “phosphorylation code,” which acts as a signaling mechanism in all eukaryotic transcriptional processes. However, the ability of these specific post-translational modifications to convey molecular signals through structural changes remains unresolved. Another challenge is that most eukaryotic protein kinases share structural similarities [3]. Our group has begun to address this question through theoretical analysis and modulation via local unconventional non-covalent interactions to examine the conformations of helical Tyr, Ser, Thr, and Pro residues in RNA Pol II CTD during transcription.

In general, a driving force is necessary for every chemical reaction. In phosphorylations, the phosphoryl group (PO_3_^−^) is transferred, for instance, from ATP to various substrates. Nucleotides transfer from ATP or guanosine triphosphate (GTP) during the synthesis of ribonucleic acids or deoxyribonucleic acids, utilizing the favorable properties of phosphoric anhydrides [30,31]. Living systems must exhibit longevity, particularly regarding genetic material. The stability of the phosphate ester bond renders it an ideal switch in information and energy transfer processes. The negative charge of the phosphate reduces the rate of nucleophilic attack on the ester, making it relatively resistant to hydrolysis and thereby enhancing DNA stability (Figure 1A). This is why phosphates persist in an aqueous environment. Moreover, they are thermodynamically unstable and can drive chemical processes to completion in the presence of a suitable catalyst (enzyme) [32].

In PO_3_^−^-transfer reactions, the nucleophile aligns with the phosphorus atom and the leaving group for an in-line attack on phosphorus. Enzyme interactions can position the nucleophile in this reactive geometry [30]. In the loose transition state, the leaving group develops a substantial negative charge relative to the ground state, resulting in the near-complete formation of metaphosphate. The monomeric metaphosphate ion becomes almost free, if not entirely so [30]. Thus, the transferred PO_3_^−^ undergoes electrostatic and geometric changes [33]. Unlike the simple intermediary metabolite phosphorylation, the group that leaves during nucleotide polymerization is PP_i_. Similarly, the activation of amino acids occurs through adenylation. Only nucleoside triphosphates can release PP_i_ while retaining a phosphate for the sugar-phosphate backbone of RNA and DNA or amino acid activation.

Critical biological processes tend to occur within quantum molecular condensates [22,34,35]. These condensates can be closed, exchanging energy, open, exchanging mass and energy, or isolated when they do not exchange. Most ionized molecules are insoluble in lipids, keeping them within membranes. Phosphates, conversely, are ionized at physiological pH and, therefore, are trapped within cells. Moreover, the electrostatic interaction of positive and negative charges constitutes the most straightforward and possibly primitive interaction between molecules. The negative charges on phosphates are essential in binding coenzymes to enzymes and “packaging” nucleic acids.

In eukaryotic cells, phosphorylation primarily occurs on three hydroxyl (OH)-containing amino acids: Ser, Thr, and Tyr, with Ser being the predominant target. Proteomic analysis has shown that phosphoserine (pSer), phosphothreonine (pThr), and phosphotyrosine (pTyr) represent 86.4%, 11.8%, and 1.8%, respectively. The fully sequenced human genome is believed to contain 518 putative protein kinases that can be classified into two families: 90 Tyr kinases and 428 Ser/Thr kinases [3,4].

#### 3.1.2. Model of Phosphorylation-Inducing Interactions Between RNA Pol II and DNA

Protein kinase enzymes catalyze the transphosphorylation of OH-containing substrates by converting ATP to adenosine diphosphate (ADP), a process facilitated by a divalent cation (typically Mg^2+^) in a substitution reaction centered around phosphorus [36]. CTD Pol II is flexible and highly disordered. Most phosphorylation sites in proteins are located in their disordered or dynamic regions. Tyr phosphorylation has emerged as a key theme in cellular regulation, primarily due to the finding that activating a Tyr kinase is the initial event in many signaling pathways. The oncogenes of various retroviruses encode Tyr kinases, which promote unregulated Tyr kinase activities that transform cells [31,37,38]. The effects of Tyr kinases are partly mediated through a substantial quantitative increase in the phosphorylation of aliphatic hydroxy-amino acids such as Ser and Thr [36]. Tyr kinases indirectly activate Ser/Thr kinases through phospholipid-derived second messengers [31]. This may explain why Tyr-phosphorylated proteins remain a small minority.

Tyr’s phenol moiety contains a hydrophilic OH group that can act as an H-bond donor or acceptor and a hydrophobic benzene ring. Thus, it exhibits a bilateral mode of interaction. A Tyr aromatic ring can form C−H⋅⋅⋅
π
 interactions, changing the potential dispersion [39]. Non-covalent interactions involving the Tyr phenol and other side-chain aromatic rings of amino acids have also been reported [40]. The structure-stability relationship of the aromatic ring stacking and the effect of the OH group phosphorylation on the stacking is an example of the “stacked form-unstacked form” equilibrium regulation [41]. It may serve as a chemical model of the biological processes involving Tyr phosphorylation.

In the stacked structure, the carbons ortho to the OH group of Tyr-OH conformation would allow the electron flow with the OH group, increasing the ring electron density. The stacking between rings with similar electron densities should have a weaker stabilizing effect [41]. Radicals play an essential role in many chemical reactions because of their unpaired single electron, which makes them highly reactive, thus quickly attacking the substrate, destroying the stable electronic structure of the substrate by snatching electrons, and making them easily reaction-activated [42,43]. Radicals are even more critical for highly polymerized organic molecules because their chain structure may provide continuous electron transfer (ET) for the reaction in thermodynamics and chemistry.

An approach to the interaction through the OH leads us to the ET theory. ET represents the tunneling transition process of an electron between two localized states on the donor and acceptor molecular groups [44]. Biological ET is efficient, specific, and frequently reversible. It has evolved from the early ideas of bridge-mediated electron tunneling in the 1960s to a quantum mechanical process [43]. However, any analytic ET theory results from some simplifying assumptions. A single-ET (SET) reaction has been proposed for the ligand–protein interaction between some molecules and Tyr residue [45,46]. The proton-coupled electron transfer (PCET) and H atom transfer (HAT) are the other two mechanisms by which the OH radical initiates the interaction with bimolecular compounds [47].

Based on these findings, we theorize that Tyr_1_ in the RNA Pol II CTD initiates reading the quantum information contained in the first base pair during transcription. The primary catalytic apparatus consists of a conserved Tyr_1_ and deoxyribose. The conserved Tyr_1_ residue performs a nucleophilic attack on the ATP phosphate moiety to release H^+^ and form a covalent pTyr–nitrogenous base ribose complex. Non-covalent interactions may influence the formation of coordination compounds [48]. The OH radical in Tyr_1_ targets the gamma-phosphorus group in the ATP chain. Then, it is transferred from ATP to Tyr_1_, enhancing the reactivity of the acceptor molecule (Figure 1B).

Figure 2 focuses on phosphorus-centered radicals. It explains that the first step is to induce a reaction in which the pTyr1-PO_4_ radical attaches to the H atom at the 2′ carbon (C) position. A deoxyribose residue in the DNA backbone has seven H atoms attached to C that are, in principle, available for abstraction by an oxidizing agent. Although DNA presents a conformational challenge, geometry optimization and orientation suggest that 2′C-H is the most probable option. After removing the deoxyribose’s H atom, the unpaired single electron becomes localized in the 2′C, increasing the electron spin density at this atom. As a result of the ET processes, the hybridization and bonding characteristics of the C atom will also change accordingly. In this scenario, the single bond between the single-electron 2′C atom and its adjacent C (1′C) atom tends to form a double bond through an electron rearrangement, accompanied by bond breaking. Thus, the single electron is transferred along the chain until the nitrogenous base is reached. The ET-specific mechanism can also be manifested as a spin-transfer reaction. The negatively charged phosphoryl group is attached to atom 2′C of the ribose. The presence of a single electron activates the C–C bond. Additionally, we hypothesize that the geometry adopted by Tyr facilitates non-covalent interactions between Tyr’s benzene ring and the aromatic ring of the nitrogenous base attached to the sugar.

The Ser_2_ residue then carries out a second nucleophilic attack. Several proteins have α-helical segments containing Ser and Thr residues. These sequences are abundant in the human proteome, particularly in IDPs/intrinsically disordered regions (IDRs) (IDPs/IDRs). Ser and Thr contain polar OH groups, with their side chains differing only by an additional methyl group for Thr. They also play similar roles in biology by forming H-bonding networks, participating in enzymatic active sites, and acting as phosphorylation targets. Because of their H-bonding potential, these residues might form H-bonding to the carbonyl oxygen in the preceding turn of the α-helix, changing its conformation [49]. xn-H_2_O fragment ions occur exclusively at Ser and Thr residues [50]. Neutral water loss by Radical-directed dissociation (RDD) is common in both amino acids, behaving similarly and facilitating nearly identical dissociation products [51]. RDD is a fragmentation technique that utilizes the collisional activation of a radical precursor to access fragmentation pathways [51].

Eukaryotic proteins have notable linker regions that are often significantly disordered. Ser and Pro are among the most disorder-promoting residues. Specifically, Ser is prevalent in IDRs and ranks as the third most disorder-promoting residue [52]. Pro is also familiar with linker regions that connect domains. On average, eukaryotic proteins consist of 450 residues, with 32% classified as disordered, amounting to an average of 145 disordered residues [53]. This suggests that disordered eukaryotic protein residues contribute to increased linker regions’ disorders. When interacting with specific binding partners, IDPs/IDRs undergo a disorder-to-order conformational transition [54]. Pro residues are crucial in modulating local and global protein structures, as Pro is the only canonical amino acid capable of forming tertiary amide peptide bonds [55,56]. In the absence of amide hydrogen, Pro can induce disorder by disrupting H-bonded secondary structures and promoting order. The trans/cis-amide isomerism at Pro creates significant pivots in the peptide backbone. Phosphorylation of Ser-Pro stabilizes the ground state of both trans- and cis-Pro amide conformations. It is proposed that phosphorylation of Ser-Thr-Pro sequences stabilizes both Pro amide isomers, although this typically occurs only in the trans-Pro conformation [57]. Furthermore, Pro is unique in non-covalent interactions; its cyclic structure results in intense inductive polarization of its C-H bonds. These bonds are potent electron acceptors and have consistently been observed to engage in C-H/O interactions in various small-molecule and protein crystal structures [58].

Biomolecular condensates formed through the phase separation of IDPs/IDRs and nucleic acids are associated with water. Water makes up about 60–70% of the condensate volume and is believed to influence the interactions between chains and chain solvent, modulating the dynamics of the condensates [59]. The active sites of many redox-active proteins are situated sufficiently close to highly polar water, allowing for a significant water component in the medium reorganization energy [44]. Electron hopping in chains of biological redox cofactors often occurs with nearly zero reaction-free energy. The influence of water molecules on the mobility and reactivity of OH radicals presents a crucial knowledge gap in our current understanding of essential reactions. However, the role of water in condensates remains unclear, particularly concerning the key functions of protein hydration water in driving phase separation. Specifically, the relevant molecular mechanisms linked to the mobility of OH radicals and the potential for diffusion in water via an H-transfer reaction remain unresolved questions [60]. Dehydrogenative transformations via a SET produce more functionalized molecules by forming C-heteroatom and heteroatom–heteroatom bonds. These reactions are characterized by chemical bond formation, including C–C, C–N, C–O, and other C-heteroatom bond formations [61].

In Figure 2, pTyr joins the CTD IDR chain with the DNA nitrogenous base through a non-covalent interaction with the deoxyribose. In contrast, the phosphorylation of Ser/Thr joins the Tyr-nitrogenous base complex with water molecules. In the next section, we will address the relationship between phosphorylation during transcription and the water associated with DNA in forming the biomolecular condensate, facilitating the quantum transmission of the genetically encoded information.

### 3.2. Water Molecules in DNA and Intrinsically Disordered Proteins

Hydration significantly influences DNA binding with proteins and ligands and the intercalation of foreign molecules. Water molecules create hydration layers around DNA, stabilizing an entangled structure [62], though they may not uniformly cover its surface. Some water molecules, particularly those in the inner hydration shells of the minor and major grooves of DNA, are tightly bound to the DNA surface. In contrast, those found in the outer shells or interacting with Na^+^ counterions are only loosely bound. The counterions neutralize the DNA’s negative charge and affect the structural network of H-bonds around DNA atomic groups, influencing the dynamic characteristics of water molecules. Water molecules exhibit more restrictions in the minor and major grooves than in free water. This phenomenon has been described as a chiral helix of water molecules in the DNA minor groove [63,64].

The dynamics of water molecules are typically described in terms of residence time, reorientation time, and vibrational frequency [65]. Water positioned near the DNA surface displays high frequencies. The changes in the characteristic residence time depend on their localization around the double helix. In the inner regions, it increases several times, reaching approximately 100 ps in the major groove and around one ns in the minor groove [65]. The H-bond reorientation between different partner atoms occurs within this time range.

Additionally, water molecules can experience translational vibrations during residence times before reorienting and transitioning to another equilibrium state. These vibrations are prominent in the low-frequency spectral range. Six modes describe the intermolecular vibrations of water molecules. DNA’s low-frequency spectral range includes the double helix’s conformational vibrations [64]. These modes consist of the stretching vibrations of H-bonds in nucleotide pairs, inter-nucleoside vibrations (near 80 cm^−1^), and backbone vibrations (near 15 cm^−1^). Translational vibrations of the hydration spine in the minor groove of the double helix occur within the frequency range of 160–210 cm^−1^ and are significantly influenced by the nucleotide sequence. The dynamics of counterions, which neutralize the negatively charged phosphate groups of the DNA backbone, are characterized by specific modes of ion-phosphate vibrations in the range of 100 to 180 cm^−1^, depending on the type of counterion [6]. The conformational vibrations of DNA and counterions are coupled with the vibrational dynamics of water molecules surrounding the double helix.

Compared to the folded globular proteins involved in processes like folding, molecular recognition, and enzyme function, the properties of water molecules on the surfaces of IDPs/IDRs remain poorly understood. Research indicates that water motions are more restricted, and the coupling between the protein and its hydration water molecules is stronger for IDPs/IDRs than for folded proteins [66]. Additionally, studies show that the diffusive water dynamics on the surfaces of IDPs/IDRs are more heterogeneous. Furthermore, the relationship between the translational diffusion of water molecules on protein surfaces and the temperature-dependent activation of large-scale protein motions is independent of the protein-folding state [67,68]. Strong electrostatic repulsions, arising from a higher net charge and a lack of compaction-inducing force due to low mean hydrophobicity, are the primary reasons for the unfolded, extended structure of IDPs/IDRs.

Having IDRs rich in amino acids with OH groups susceptible to phosphorylation as a post-translational modification offers advantages compared to the globular structure of proteins. The general phosphorylation reaction scheme is represented in:MgATP^−^ + substrate-O: H → substrate-O: PO_3_^2−^ + MgADP + H^+^

The main proton transfer mechanism is believed to be an Eigen–Zundel–Eigen (EZE) process, where a “distorted” Eigen cation is the most stable form, and the Zundel cation primarily acts as an intermediate complex [69,70,71,72]. Our model illustrating the formation of the tetrahedral structure of water is presented in Figure 3. The release of a proton of H^+^ during the phosphorylation pulse modifies the nearby water molecules. The kinetic energy the proton gains as it is transformed into a free particle is employed to attract one water molecule to form the hydronium ion (H_3_O^+^). The H_3_O^+^ positive charge attracts two additional water molecules and forms the tetrahedral structure H_7_O_3_. It features a bifurcated H-bond that links to two neighboring water molecules and is distinctive, representing its minimum energy structure. The predominance of positive charges binds the water molecules together, probably contributing to what has been observed in other studies about the helical structure of water. This will be addressed later in the Section 4.

#### 3.2.1. Physical Model of Phosphorylation Mediating H^+^ Proton Transfer

ATP hydrolysis is a major source of H^+^. The process involves removing the terminal phosphate from ATP to produce ADP, releasing free energy and P_i_ simultaneously, which requires water as an additional substrate. Phosphotransfer reactions are another critical source of H^+^. Each protein kinase catalyzes the enzymatic transfer of a phosphate group from ATP. In eukaryotes, it covalently attaches to one of the three amino acids containing hydroxyl groups: serine, threonine, or tyrosine. This section will address the physical model of phosphorylation that induces the tetrahedral water structure through H^+^ proton transfer, as illustrated in Figure 3. The released H^+^ proton acts as a free particle. The quantum-mechanical motion of a free particle is analyzed in the Hamiltonian formalism, and its kinetic energy is calculated by solving the Schrödinger equation:
(1)
$$\hat{H}\left|\left.\psi \right\rangle =\right.E\left|\left.\psi \right\rangle \right.$$

where 
$\left|\left.\psi \right\rangle \right.$
 is the wave function of a free particle:
(2)
$$\left|\left.\psi \left(\vec{r}\right)\right\rangle =Ae^{i\left(\bar{k}.\bar{r}\right)}\right.$$

where 
${\hat{H}}_{p}=\hat{K}+\hat{V}\left(\bar{r}\right)$
, and 
$\hat{K}=\frac{-\hbar^{2}\nabla^{2}}{2m_{p}}$
, 
$\hat{V}\left(\bar{r}\right)=0$
 because it is a free particle. Then, Equation (1):
(3)
$$\left[-\frac{\hbar^{2}\nabla^{2}}{2m_{p}}+\hat{V}\left(\bar{r}\right)\right]\left|\left.\psi \left(\vec{r}\right)\right\rangle \right.=E\left|\left.\psi \left(\vec{r}\right)\right\rangle \right.$$


Then, replace (2) in (3):
$$-\frac{\hbar^{2}\nabla^{2}}{2m_{p}}\left(Ae^{i\left(\bar{k}.\bar{r}\right)}\right)=EAe^{i\left(\bar{k}.\bar{r}\right)}$$


We obtain that:
$$E=\frac{\hbar^{2}k^{2}}{2m_{p}}$$


Considering 
$\nabla^{2}=-\frac{\partial^{2}}{{\partial \vec{r}}^{2}}$


Water is a complicated quantum problem of eight electrons and three nuclei (two of H and one of oxygen). The Hamiltonian of a water molecule would be expressed:
$${\hat{H}}_{H_{2}O}={\hat{K}}_{O}+{\hat{K}}_{2H}+{\hat{V}}_{OH}+{\hat{V}}_{8e}$$


H^+^ does not state freely. It is highly reactive and is quickly solvated by surrounding water molecules. H_3_O^+^ is the only molecular cation that forms in water with H^+^ ions. An electron from the lone pair of a 
2p
 orbital of oxygen jumps to the 
1s
 orbital of H^+^, leaving a hole in a 
2p
 orbital of oxygen. Thus, when the proton approaches the H_2_O molecule, the H_3_O^+^ is formed, which is electropositive with a Hamiltonian:
$${\hat{H}}_{H_{3}O}={\hat{H}}_{p}+{\hat{H}}_{H_{2}O}$$


The H_3_O^+^ has a trigonal pyramidal molecular geometry with the oxygen atom at its apex. The H^+^ ion utilizes its free energy to induce the formation of water clusters, since the H_3_O^+^ cation attracts two additional water molecules, H_2_O, to create a regular tetrahedron with the following Hamiltonian 
$\left({\hat{H}}_{T}\right)$
:
$${\hat{H}}_{T}={\hat{H}}_{H_{3}O}+2{\hat{H}}_{H_{2}O}+{\hat{V}}_{H_{3}O-H_{2}O}$$


When the H_3_O^+^ and OH^−^ ions come close enough, a water wire continuously connecting them experiences collective compression during their recombination and creates a current. In the next section, we will explain how the supercurrent is created using the model of electron correlation in oscillating resonant quantum states. The process starts when the oxygen of the H_3_O^+^ shares a lone pair electron with the H^+^ proton, leaving a hole in the 
2p
 orbital. If an electron located in a 
2p
 orbital moves to a 
1s
 orbital, energy is released in the form:
$$E_{2p}-E_{1s}=\hbar \omega$$


This energy is used to correlate electron pairs in the 2
p 
orbitals of the oxygens, resulting in a superconducting current inside the tetrahedron.

#### 3.2.2. Superconductor Character of Water Induced by H^+^ Proton Transfer

The movement of an H^+^ proton during phosphorylation reactions induces an electric current of electrons into the water clusters. In this work, we explain that the conductivity of water is not precisely due to the PT theory, but to the supercurrent created by the delocalized motion of electrons moving in resonant quantum states. We prefer to call this phenomenon a supercurrent induced by a proton. The kinetic energy that the proton brings when it becomes a free particle is used to attract a water molecule. Still, when the H_3_O^+^ is formed, an electron from the lone pair of a 
2p
 orbital of oxygen jumps to the 
1s
 orbital of H^+^, it releases a Boson of quantum energy, inducing the subsequent electron pairs correlation and movement.

Previously, we described the general oscillatory resonant quantum states process between electron and hole pairs forming in 
π
-orbitals. The overlap of two 
p
_z_ orbitals brings two electrons with equal spin closer, allowing them to experiment with two interactions: the Coulombian repulsion and the electron-vibrational energy-electron. The Coulombian repulsion forced electron one (
e1
) to occupy the position of hole one 
h1
. During the translation, 
e1
 emits a quantum of energy 
ℏω
 to electron two (
e2
). The energy absorbed is used by 
e2
 to occupy the hole two (
h2
) position. In the second half of the oscillation, 
e1
 and 
e2
 exchange 
ℏω
 to return to their original positions. The energy difference between the electrons is the same as 
ℏω
 [10]. That is the condition for the electron and hole pairs, one below and one above the Fermi level with opposite momentum, to oscillate. The electrons cannot occupy other states, so they do not interact with other atoms in the molecule. The resistance or dispersion energy is canceled.

In Figure 4A, we explain the process of superconducting water based on these same principles. When the H_3_O^+^ cation is formed, the kinetic energy of the H^+^ proton attracts an electron from a lone pair located in a free orbital 
2p
 of an oxygen (O_1_) atom to the empty 
1s
 orbital, leaving a hole. The displacement of this electron then releases energy 
ℏω
 and leaves a hole in its position. This 
ℏω
 is now used by another electron located in the 
2p
 orbital of the second oxygen (O_2_) to move to the new hole left.

#### 3.2.3. Application of the Physical-Mathematical Model for the Correlation of Pairs of the Same Type of Fermions (Two Electrons) That Exchange a Boson to Explain Superconductivity in Water

In Figure 4B, the correlation of pairs of electrons that exchange a boson explains the superconductivity of water. In the first part, the electron with down spin (
↓
) moves to the empty 
1s
 orbital of H^+^. In the second, the processes of formation of correlated pairs are represented when the electron that moves to the 
1s
 orbital of H^+^ is the one that has spin up (
↑
). The electrons that correlate to form pairs are 
e2
 and 
e5
 because they have opposite momentum, spin, and different energies. Their energy difference equals that of the boson (photon) they exchange. Electrons 
e3
 and 
e4
 cannot pair because when exchanging the boson (photon), they do not have a permissible space (hole) for the oscillatory movement to occur in the resonant quantum states. Electrons 
e1
 and 
e4
 also cannot oscillate in resonant quantum states because they involve transitions of electrons within the same orbital.

Two processes are required to form electron pairs. Process one involves a half oscillation. In this half oscillation, we go from state a to state b. In process two, the fermion pairs exchange the boson, performing a second oscillation that defines the oscillatory resonant quantum states. In this second-half oscillation, the system goes from state b to c, where the initial state is reached. The internal Hamiltonian (
$H_{INT})$
 of the system in the state a (
$H_{INT}^{a}$
) is given by:
$$H_{INT}^{a}=U_{e_{2}e_{5}}+V_{e_{2}Fe_{5}}$$

where 
$U_{e_{2}e_{5}}$
 is the Colombian repulsion potential between electrons 
e2
 and 
e5 
and 
$V_{e_{2}Fe_{5}}$
 is the interaction potential between 
e2
 that emits a boson and 
e5
 that absorbs it.

The 
$H_{INT}$
 of the system in the state *b* (
$H_{INT}^{b}$
) is given by:
$$H_{INT}^{b}=U_{e_{5}e_{2}}+V_{e_{5}Fe_{2}}$$

where 
$U_{e_{5}e_{2}}$
 is the Colombian repulsion potential between electrons 
e5
 and 
e2
_,_ equal to 
e2
 and 
e5
. Then, 
$U_{e_{2}e_{5}}=U_{e_{5}e_{2}}$
. 
$V_{e_{5}Fe_{2}}$
 is the interaction potential between 
e5
_,_ which emits a boson, and 
e2
_,_ which absorbs it. Then, 
$V_{e_{2}Fe_{5}}=-V_{e_{5}Fe_{2}}$
.

The total Hamiltonian would be:
$$H_{INT}^{T}=H_{INT}^{a}H_{INT}^{b}=\left(U_{e_{2}e_{5}}+V_{e_{2}Fe_{5}}\right)\left(U_{e_{5}e_{2}}+V_{e_{5}Fe_{2}}\right)$$


Considering:


$H_{INT}^{a}H_{INT}^{b}={∆}^{2}$
, the correlation energy of the electron pair,


$V_{e_{2}Fe_{5}}=-V_{e_{5}Fe_{2}}$
 and 
$U_{e_{2}e_{5}}=U_{e_{5}e_{2}}$

$$\left(U_{e_{2}e_{5}}+V_{e_{2}Fe_{5}}\right)\left(U_{e_{5}e_{2}}+V_{e_{5}Fe_{2}}\right)=\left(U_{e_{2}e_{5}}+V_{e_{2}Fe_{5}}\right)\left(U_{e_{2}e_{5}}-V_{e_{2}Fe_{5}}\right)$$


Then, we obtain the classical scattering law of correlated pair formation in superconductivity:
$${∆}^{2}=U_{e_{2}e_{5}}^{2}-V_{e_{2}Fe_{5}}^{2}$$


Here, the existence of superconductivity in water is demonstrated.

## 4. Discussion

Chemical reactions involving the formation or cleavage of P-O bonds in phosphate esters are ubiquitous in biological systems. This work made a theoretical analysis and modulation via local unconventional non-covalent interactions to explain how the RNA Pol II CTD utilizes the 52 repeats of the sequence Tyr_1_Ser_2_Pro_3_Thr_4_Ser_5_Pro_6_Ser_7_ to interpret the genetic information encoded quantumly in the nitrogenous bases. In a prior study based on circuit quantum electrodynamics theory, our group described the fundamental structure of DNA genes as a combination of non-perturbative circuits. By considering each base pair as an RF-SQUID (A-T, T-A, G-C, C-G), the dispersive detuning regime enables the quantum measurement of the qubit state and its manipulation. A sequential series of phosphorylation reactions was proposed as the driving pulse for transcription. Irradiating at the qubit frequency coherently controls the qubit, while irradiating at the resonator frequency introduces photons into the cavity that become entangled with the qubit states [11] The coupling between *n*-phosphorylation pulses and a specific qubit could be selectively achieved through frequency matching [11].

Traditionally, non-covalent interactions, such as H-bonds, hydrophobic effects, electrostatic interactions, and van der Waals forces, are fundamental for protein folding, structural stability, and function [73]. The 
π
-moiety-involved interactions, such as 
π−π
 stacking, CH
 − π
, cation/anion
 − π
, *n* → 
π
*, and XH
 − π
 interactions (XH = NH, OH, SH), are equally crucial non-bonded interaction forces [4,74,75]. These non-covalent interactions govern photophysical properties that are sensitive to the microenvironment. Thus, by altering these interactions, selective sensing for a particular analyte can be achieved [76].

It has been reported that Tyr can form 
π − π
 stacking interactions with aromatic rings, while the side chains of Ser and Thr can create OH
 − π
 and NH
 − π
 interactions [4]. Compared to H-bonding interactions, the dispersion forces that dominate CH
 − π
 and 
π − π
 stacking interactions are relatively weak. However, dehydration incurs a much lower (or zero) energy cost. As a result, significant interaction strengths persist in the aqueous phase for both CH
 − π
 and 
π − π
 stacking interactions [77,78].

Here, we found that the phosphate group in pTyr produces H-abstraction at 2′C of deoxyribose. The distribution of the unpaired electron via SET activates the C–C bond, supported by a readjustment of the electronic configuration. Living systems use proximity to regulate biochemical processes [79], and some authors have characterized Tyr’s defining trait as a stronger adhesive [80]. Ligand–protein and protein–protein interactions via weak and transient connections are essential for regulating all dynamic biological phenomena.

Various molecular drivers influence water’s H-bonding network changes during homotypic and heterotypic phase separations [59]. Molecular characteristics, such as affinity, valence, and the competition between intermolecular and intramolecular contacts, also contribute to phase separation [81]. Phase equilibria are established by balancing species-specific chemical potentials with osmotic pressure across the coexisting phases [82,83,84]. Aromatic residues and other polar moieties play a role in the phase transitions of IPDs [84,85]. Identifying these critical and dynamic interactions remains a challenge.

The OH radical in aqueous environments plays a significant role in initiating specific reactions. In one study, the OH radical extracts an H atom from the polyacrylamide main chain. This action simultaneously triggers the SET process on the same chain, as shown by the spin-density transfer, which leads to a reduction in the dissociation energy barrier of the C–C bond from approximately 90 to about 20 kcal/mol [42]. It was found that the SET process is crucial to the mechanism initiated by the attack of radicals on specific polymers [86]. Modeling studies have predicted the presence of H-bonding and hydrophobic interactions mediated by the amino acid residues of Ser and Tyr, which elucidate odorant receptors and binding sites [39]. Based on these interactions, various heterobifunctional small molecules have recently been developed to enhance the proximity of two or more proteins, triggering proximity-dependent modifications [87].

The conversion of the Tyr phenol OH group to the phosphoester group through phosphorylation significantly decreases stability due to the hydrophilic nature of the negatively charged phosphoester moiety. Specific biological processes are regulated by modifying the Tyr side group through phosphorylation, with its observed effect on intermolecular stacking interactions suggested as the basis for understanding molecular recognition and the biological regulation of reactions [87].

RNA Pol II translocates across much of the genome, but various DNA lesions can block it. Protein-nucleic acid interactions are essential for transferring biological information and depend on sequence-specific recognition. Proper phosphorylation of the CTD is necessary for resistance to several DNA-damaging agents and the repair of specific DNA lesions [88]. Chemical modifications of nucleic acid bases, found across all domains of life, have become a key regulatory mechanism. For example, PARylation controls various processes such as chromatin remodeling and transcription activation [89]. The modifications targeting nucleic acids can be categorized into three types: ADP-ribosylation of DNA bases, ADP-ribosylation of phosphorylated DNA ends, and ADP-ribosylation of phosphorylated RNA ends [90]. PARP and PARP-like enzymes can catalyze the reversible ADP-ribosylation of phosphorylated DNA ends as an evolutionary mechanism to repair DNA lesions, promoting genome integrity. They can also ensure a surface for protein interactions [91,92].

Interactions between aromatic rings are crucial in protein and protein–DNA systems for stabilizing proteins and facilitating various regulatory processes. Insights into DNA interactions involving the side chains of amino acid residues from transcription factors and the nucleobases of DNA have significantly advanced our understanding of the roles played by weak interactions during the initial stages of DNA transcription [87]. Additionally, ring substituents affect the stability of stacked structures. One study analyzed the transcription factor HcaR from *Acinetobacter* sp. and four HcaR-ligand complex structures. Each ligand was bound to the same deep cavity, surrounded by multiple electron-rich aromatic and hydrophobic residues, including Tyr. Several hydrophilic residues, such as Ser and Thr, could also form direct or water-mediated hydrogen bonds with the ligand at the dimer interface within the cavity [93]. Intermolecular stacking interactions have been shown for nucleotides and complexes with aromatic ligands in both solution and solid states. As expected, these interactions depend on the structure of the aromatic rings and can be interpreted as resulting from the interaction between molecular orbitals [87].

C–H/O interactions occur between neutral or anionic oxygen lone pairs and highly polarized C–H bonds. These interactions are often described in electrostatic terms related to charge attraction, similar to H-bonds. While there is an electrostatic aspect to C–H/O interactions, experimental and computational analyses have shown that they are primarily stereoelectronic, characterized by the delocalization of oxygen lone pair electrons into an alkyl H antibonding orbital [94]. Due to their significant stereoelectronic component, it has also been noted that the strength of C–H/O interactions is mainly independent of electrostatics in polar environments, such as water.

The mechanism of protons moving through water is central to acid-base chemistry reactions. Developing a molecular basis for these phenomena is significant in energy conversion applications, such as designing efficient fuel cells. For a successful transfer, the model requires solvent reorganization around the proton-receiving species to establish a coordination pattern similar to the species it will convert to, a process known as pre-solvation [62,95]. Despite various characterizations of the Grotthuss mechanism, the role of the connectivity of the water network has not been emphasized.

Introduced in 1806 by Grotthuss, the concept of proton hopping is still regarded as the most efficient PT mechanism, typically implying the PT between chains of host molecules through elementary reactions involving H-bonds [96,97,98]. Popov et al. noted that PT between molecules in pure and 85% aqueous phosphoric acid occurs via surprisingly short proton jump lengths (0.5–0.7 Å), accompanied by a reduction in conductivity, contrary to the expected enhancement typically associated with the Grotthuss mechanism [99]. Agreeing with this result, in our work, we argue that the conductivity of water is not solely due to proton movement but rather to the supercurrent generated by the delocalized motion of electrons in resonant quantum states. The kinetic energy the proton gains when it becomes a free particle is exclusively used to attract a water molecule. An electron supercurrent is created by the transfer of a proton. The mutual neutralization of H_3_O^+^ and hydroxide OH^−^ ions is primarily attributed to an ET mechanism, for which a minor contribution is linked to PT [100].

H_3_O^+^ is the only molecular cation that forms in water with H^+^ ions. These cations do not exist freely; they are highly reactive and are quickly solvated by surrounding water molecules [100]. Among the over 1000 crystal structures of hydronium ions recognized by the Cambridge Crystallographic Database, the H_3_O^+^ ion with a bifurcated H-bond connecting to two adjacent water molecules is unique, representing its minimum energy structure. Huang et al. demonstrate that this optimized geometry aligns with an unusual transition state characterized by a single vibrational frequency associated with protonated water structures [101]. Some solvation structures can be pretty significant: the ion H_3_O^+^(H_2_O)_20,_ often referred to as “magic” due to its excellent stability compared to other structures with a similar number of molecules [102] would trap the proton inside a dodecahedral formation.

Water features a helical component in its structure, primarily because of the proton-hopping properties of the H_3_O^+^. The foundation for the proposed water structure consists of tetrahedral substructures that, when combined, form long helitetrahedral chains shaped like Boerdijk–Coxeter (BC) helices. The BC helix represents a linear arrangement of regular tetrahedra [103].

PT has been considered to involve the simultaneous jumps of multiple protons [104]. In this context, protons can delocalize over extensive H-bonded networks. This phenomenon occurs when water molecules form isolated chains in confined environments such as proteins. Rapid PT occurs through transient water wires forming during acid–base reactions in ice and water. When the H_3_O^+^ and OH^−^ ions come within approximately 6 Å of each other, a water wire continuously connecting them experiences collective compression during their recombination [99]. This event leads to the synchronized movement of three protons on a timescale of tens of femtoseconds. The formation of polarized water wires appears to be a crucial precursor to correlated PT events. This raises the question of whether wire-like structures can exist in liquid water. Charge migration includes bursts of activity along proton wires within the network, characterized by the coordinated movement of several protons and resting periods longer than expected, reminiscent of a jump-like diffusion mechanism. This remarkable dynamic activity is partly driven by the proton wires’ ability to undergo collective compressions [105]. Previous studies have highlighted that a wide distribution of closed rings characterizes liquid water [106].

We divide PT through water molecules into consecutive steps between adjacent multicopy donor–acceptor pairs. One proton donor is marked at the beginning, creating multiple copies. A sphere is drawn around this donor, generating numerous copies for all possible acceptors within this sphere and potential additional donors. Protons hop between the multicopy donors and acceptors. If a group of acceptor copies becomes protonated, this group is marked as the new donor for the next transport step. All non-protonated copies of this acceptor must take up a proton from one of the remaining donor copies within a specified timeframe. Subsequently, the newly marked donor becomes the center of a sphere containing multicopy groups for the next transport step. Protonated water clusters also play a crucial role in the confined spaces and cavities of proteins and biomolecules.

H^+^-transfer can be a very rapid process involving water and is significantly affected by structural fluctuations. A rough estimate of the total activation energy for the proton exchange reaction involving OH^−^ has previously been assessed at 3 kcal/mol [107]. Early electron movement is noticeable compared to the position of the H^+^ proton [108]. This suggests a hybrid mechanism incorporating electron–PT elements. A weak H-bond is formed between the OH–H and the H-bond-accepting water in the transition state. Consequently, the radical center relocates to a new site within the H-bond network [60].

We also hypothesize that the movement of protons and the transformations they trigger in the surrounding biological water alter the frequency during the pulses of sequential phosphorylations. The coherent vibrational character of the coupled solute and solvent arises when strong H-bonds exist between the solute and the water molecules [109]. The ability of Ser/Thr to form H-bonds links the CTD sequence Tyr_1_Ser_2_Pro_3_Thr_4_Ser_5_Pro_6_Ser_7_ repeats to water molecules, creating a nitrogenous base–deoxyribose–CTD–water condensate. As a result, a revision has been proposed for the typical depiction of the interaction between DNA proteins and the solvating water molecules, treating their interaction as a perturbation. Especially in the presence of strong H-bonding between the solute and water molecules, water vibrations tend to be delocalized [110], suggesting that the vibrational eigenstates associated with the solute in water may also be delocalized, with correlations extending throughout its solvation shell. Thus, the extent of the coherent vibrational character of the coupled solute and solvent determines the distance scale and mechanism over which concerted bimolecular chemical reactions, such as PT, can occur. In aqueous chemical reactions, these vibrational motions and dissipation would also act as the source of the detuning processes.

This is a theoretical approach to explain the link between RNA Poll II CTD phosphorylation and the formation of biological transcriptional condensates, highlighting the role of phosphates and their interactions with water. Although recent years have seen progress in experimental and computational models that enhance our understanding of biological macromolecules, several limitations still exist because these models rely on solving an ensemble of closely related structures, each with a similar likelihood of occurring under experimental conditions. Challenges in analyzing these macromolecular entities include their inherent flexibility. Moreover, the target macromolecule may not always be suitable for these experiments, or the experimentally observed conformation cannot be directly used for docking studies. Sometimes, the necessary insights into the binding mechanism are unavailable because the ligand–receptor complex structure is missing or lacks time resolution. Additionally, these methods usually only record the final stage of binding, offering limited or no details about the intermediate steps involved in the process. Since the theoretical framework in research offers a foundational understanding of the study’s concepts and relationships, our findings reinforce the need to strengthen theoretical models for an accurate approach to structural relationships.

## 5. Conclusions

Ser-Thr-Pro sequences are plentiful in the human proteome, particularly in IDPs/IDRs. This abundance highlights their biological significance as dynamic sites for structural modulation. The human RNA Pol II CTD comprises 52 disordered Tyr_1_Ser_2_Pro_3_Thr_4_Ser_5_Pro_6_Ser_7_ heptad repeats. The dynamic phosphorylation of Ser_2_, Thr_4_, and/or Ser_5_ within the Pol II CTD establishes a “phosphorylation code” as a signaling mechanism in all eukaryotic transcriptional elongation and termination activities.

Protein residues may be essential components of PT in H-bond networks. Negatively charged phosphates react under the catalysis of enzymes that couple PT with electron transfer. Partially replacing H in deoxyribose with electron-rich PO_3_^−^ enhances strong electronic interactions between Tyr and the sugar, linking the aromatic ring of Tyr with the aromatic nucleobases of DNA. Subsequently, the phosphorylation of Ser-Thr-Pro sequences alters the secondary structure and establishes H-bond connections with water molecules. The results suggest that this strategy can significantly increase the vacancy formation energy and the migration energy barrier of PO_3_^−^.

Meanwhile, the H^+^ ion utilizes its free energy to form H_7_O_3_ water clusters. The vibrational dynamics of water in aqueous chemical reactivity indicate that this reaction system likely involves an H-bond between the water molecule and the solute (CTD sequence Tyr_1_Ser_2_Pro_3_Thr_4_Ser_5_Pro_6_Ser_7_). The concerted reaction establishes a framework of DNA–protein–water interactions. Additionally, we anticipate that the system exhibits coherent vibrational motions between the coupled solute and solvent, reflected in the modulation of H during the concerted reaction, as the reaction system involves oscillations in proton donor–acceptor distances.

This natural electronic confinement-restrained strategy ensures the optimal temperature and pressure conditions needed to teleport the quantum information of each base pair. Utilizing the confinement effect of electron interactions offers new perspectives for exploiting and overcoming high-energy-density materials. Additionally, incorporating PO_3_^−^ increased the number of lone pairs of electrons and optimized the band structure, benefiting from suitable conduction bands, sufficient electrons, and effective ET. The system limited the formation of intermediate products and enhanced catalytic efficiency. Multi-electron reduction achieved in electron-rich catalysts with high local electron concentrations will reduce the production of undesirable intermediates. With the active involvement of kinases and phosphatases, the process is reversibly converted to facilitate the reversible participation of multiple electrons in the electrochemical transcriptional process.

Also, in this work, we describe the superconducting character of water through the correlation of two electrons that exchange a boson. The coexistence of empty orbitals and outer lone pairs is beneficial, as the empty orbitals can accept electrons, acting like holes. We have developed this theoretical model more extensively in our previous works, and it is a universal model for explaining superconductivity. But unlike these earlier works, the initial energy here is provided by the movement of a free H^+^, creating a proton-driven supercurrent of electrons. Nature chose phosphates as transcription machinery-driven forces because of their multi-ET capabilities and high energy efficiency.

## Figures and Tables

**Figure 1 cimb-47-00571-f001:**
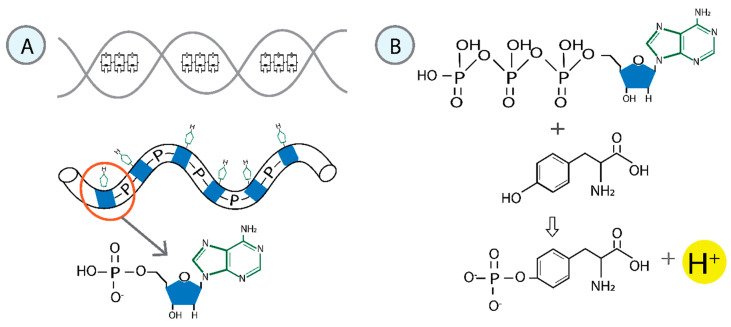
The figure illustrates the fundamental structure of DNA and ATP. (**A**) The primary structure of DNA is examined as a combination of quantum circuits. Each base pair serves as a qubit, functioning like an RF-SQUID. The sugar-phosphate backbone resembles a transmission line with distributed parameters [11]. (**B**) Chemical description of ATP. The amino acid tyrosine contains a phenol group in its side chain that can undergo phosphorylation as a post-translational modification. The transphosphorylation of OH-containing substrates is a substitution reaction centered around phosphorus and releases a hydrogen proton. The beta and gamma phosphate groups are transferred from ATP to tyrosine.

**Figure 2 cimb-47-00571-f002:**
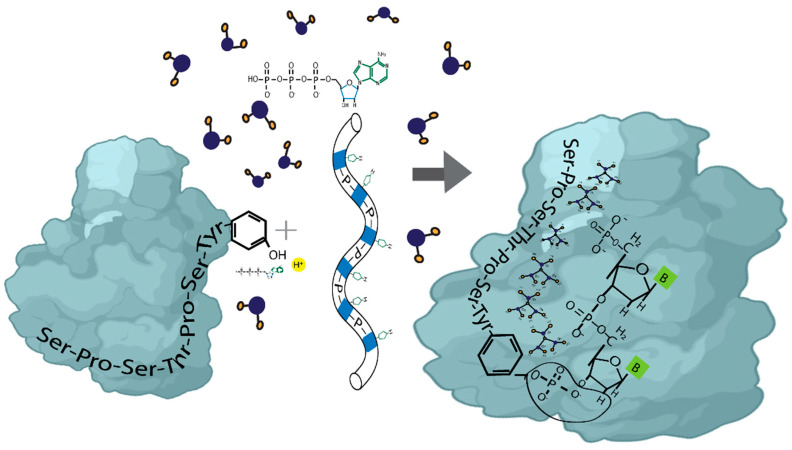
Formation of the RNA polymerase II-DNA-water molecular condensate. The initial step involves inducing a reaction that attaches the phosphorylated tyrosine to the sugar of a nitrogenous base. After removing the hydrogen atom from deoxyribose, the unpaired single electron becomes localized at the corresponding carbon (2′C). The geometry adopted by tyrosine promotes non-covalent interactions between the protein and the nitrogenous base. A sequential series of phosphorylations is initiated, modifying the structure of the water associated with the molecular condensate. These non-covalent interactions govern the photophysical properties of the system.

**Figure 3 cimb-47-00571-f003:**
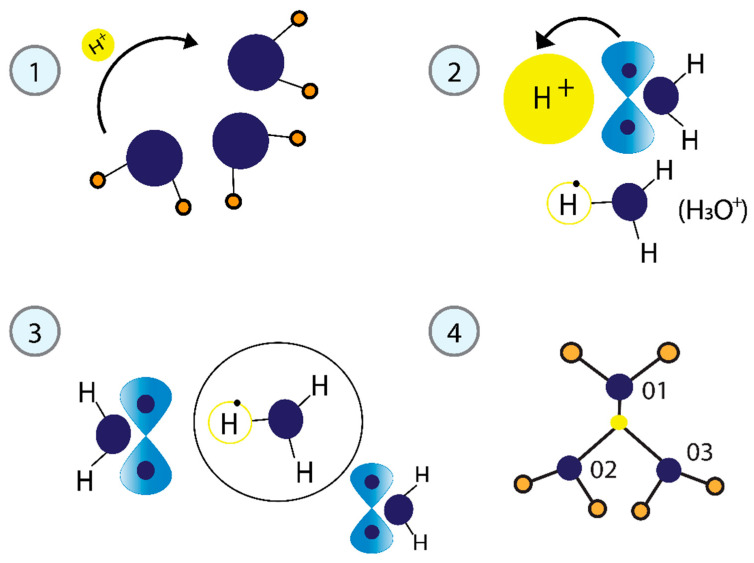
The process of forming the tetrahedral structure of water. (**1**) The released proton (H^+^) functions as a free particle. (**2**) The kinetic energy generates the H_3_O^+^ cation from one water molecule. (**3**) H_3_O^+^ is highly reactive and is quickly solvated by neighboring water molecules. (**4**) The H_3_O^+^ ion, linked by a bifurcated H-bond to two adjacent water molecules, represents the minimum energy structure.

**Figure 4 cimb-47-00571-f004:**
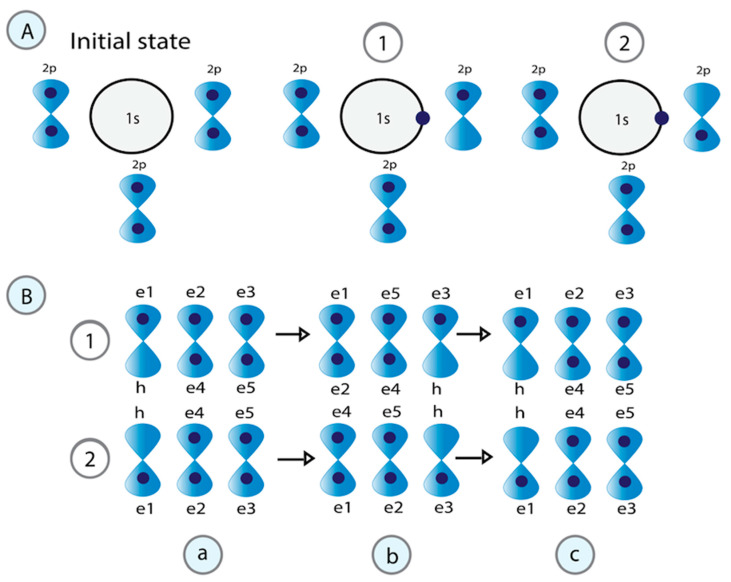
Structure of water associated with DNA in the molecular condensate. (**A**) During the formation of the H_3_O^+^ cation, one electron from a lone pair in the oxygen orbital 
2p
 moves to the empty H^+^
1s
 orbital, creating a vacancy. The released energy (
ℏω
) is used to form correlated pairs of electrons. The electrons that correlate to form pairs should have opposite momentum, spin, and different energies. Their energy difference equals that 
ℏω
. (**B**) Two processes are required to create electron pairs. Processes a to b involve a half-oscillation. In processes b to c, the fermion pairs exchange 
ℏω
, and a second oscillation defines the oscillatory resonant quantum states. The difference between 1 and 2 is the electron’s spin that leaves the vacancy.

## Data Availability

All data are included in the manuscript.

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
