# Peer review of "The Transcription Machinery and the Driving Force of the Transcriptional Molecular Condensate: The Role of Phosphates"

_cimb, 2025, doi:10.3390/cimb47070571_

Round 1
Reviewer 1 Report
Comments and Suggestions for Authors
The authors of this study present a theoretical framework in which the role of the phosphate group in the formation of transcriptional molecular condensates comprising RNA Polymerase II, DNA, and water molecules is modeled using quantum circuit elements such as Josephson junctions and SQUIDs. This approach introduces an innovative perspective on the mechanistic underpinnings of DNA transcription by drawing analogies from quantum information theory and circuit quantum electrodynamics. However, the framework remains entirely hypothetical, as it lacks experimental validation or supporting biophysical simulations. The manuscript would be significantly strengthened if the authors provided experimental evidence to support their theoretical claims. There is no experimental or theoretical evidence supporting superconductivity in hydrated proton clusters under ambient or even low-temperature aqueous conditions.
I have other critical comments in this article as follows.
My other concerns are as below.
- Sections 3.1.1 (Phosphates as the driving force), 3.1.2 (Model of phosphorylation inducing interactions between RNA Pol II and DNA), and 3.2 (Water molecules in DNA and intrinsically disordered proteins) do not present any original results generated by the authors in this study. Instead, these sections largely consist of summaries and interpretations of previously published work. As such, their placement within the Results section is questionable. Without novel data, simulations, or computational outputs, these discussions would be more appropriately positioned in the introduction or discussion sections, where background and conceptual frameworks are typically addressed.
- In lines 245-246, the authors mention “a conformational challenge, geometry optimization and orientation suggest that 2'C-H is the most probable option” What was the rational for this? Is it based on the previous studies or what? In principle, oxidizing agents preferentially abstracts hydrogens from the more solvent-accessible positions C5′ and C4′ with C3′ and C2′ being less favorable. So, removing the H atom from the 2'C cannot be agreed upon.
- In Line 374, the authors proposed the formation of the Eigen cation (H₉O₄⁺) or tetrahedral cation. But Zundel cation (H₅O₂⁺) is equally or even more realistic in the protein environment. The authors should shed some light on it and explain their rationale for choosing the Eigen cation (H₉O₄⁺) over the Zundel cation.
- Defining atomic orbitals (1s, 2p) in a molecular environment like H₂O or H₃O⁺ is misleading in line 423.
- Fundamentally, I cannot agree with the author with treating released protons in the phosphorylation as the free particle in line 396. In real biological or aqueous environments, protons are never free particles; hence, 𝑉̂(𝑟̅)=0 cannot be justified. Proton transfer occurs through a complex potential surface shaped by the electric field of nearby water molecules, proteins, and phosphate groups, etc.
After carefully reviewing the entire article, the main objective or practical significance of the study remains unclear. The authors propose that water within transcriptional molecular condensates comprising RNA Polymerase II, DNA, and water exhibits superconducting properties. However, the central concern is the validity of this claim. Specifically, the manuscript does not provide experimental evidence to support the existence of such superconducting behavior in biological systems. Without empirical validation or mechanistic grounding, the hypothesis remains speculative and difficult to assess within the context of current molecular biology.
Author Response
Dear Editor,
We are very grateful for all your comments and suggestions; they significantly improve the quality of the paper.

Reviewer 2 Report
Comments and Suggestions for Authors
This article mainly explores the key mechanism of phosphorylation in the transcription process. Research has found that the carboxyl terminal domain (CTD) of RNA polymerase II dynamically phosphorylates to form a "phosphorylation code" that regulates gene transcription. The article proposes an innovative quantum model that treats genes as quantum circuits, DNA base pairs as quantum bits, and reads genetic information through phosphorylation pulses. The study also revealed the phosphorylation mediated proton transfer mechanism, discovering that water molecules form ordered structures around DNA, and proton transfer may induce superconductivity. These findings provide a new perspective for understanding transcriptional regulation, particularly how phosphorylation promotes genetic information transmission through quantum effects.
Specific comments are below.
- Some theoretical models have not been experimentally validated and lack direct experimental evidence to support them. Nevertheless, the viewpoint presented in this article still holds significant importance, and it is recommended to find corresponding experimental evidence in future research.
- The Introduction section suggests adding some background knowledge and explanations of basic concepts appropriately, so that a wider readership can understand and appreciate this research.
- Part 3.2.1: Provide detailed explanations and clarifications of key calculation processes and assumptions.
- It is suggested to discuss in the discussion section the effects of other DNA modifications on protein function, such as PARylation, whether it affects protein function by affecting phosphorylation or polarity (Reference resources: PMID: 39939798).
- The discussion section of the paper lacks sufficient exploration of the limitations of the research. Although the author has raised some potential challenges and issues, they have not fully revealed the limitations of the research, such as the differences between theoretical models and actual situations, and the difficulty of experimental verification.
- Please check the format of reference, eg. Line 549, reference 74 75,76, missed “,”;
- Line 523, (Riera Aroche, Ortiz García, Sánchez Moreno, et al., 2024) is reference or not? If it is a reference, please modify the format.
Author Response

(The authors gave the same response as above.)

Round 2
Reviewer 1 Report
Comments and Suggestions for Authors
The authors modified the manuscript to address my concerns.